# Mia40 is a trans-site receptor that drives protein import into the mitochondrial intermembrane space by hydrophobic substrate binding

Valentina Peleh[1], Emmanuelle Cordat[2], Johannes M Herrmann[1]*

[1]Cell Biology, University of Kaiserslautern, Kaiserslautern, Germany; [2]Department of Physiology, University of Alberta, Edmonton, Canada

**Abstract** Many proteins of the mitochondrial IMS contain conserved cysteines that are oxidized to disulfide bonds during their import. The conserved IMS protein Mia40 is essential for the oxidation and import of these proteins. Mia40 consists of two functional elements: an N-terminal cysteine-proline-cysteine motif conferring substrate oxidation, and a C-terminal hydrophobic pocket for substrate binding. In this study, we generated yeast mutants to dissect both Mia40 activities genetically and biochemically. Thereby we show that the substrate-binding domain of Mia40 is both necessary and sufficient to promote protein import, indicating that trapping by Mia40 drives protein translocation. An oxidase-deficient Mia40 mutant is inviable, but can be partially rescued by the addition of the chemical oxidant diamide. Our results indicate that Mia40 predominantly serves as a trans-site receptor of mitochondria that binds incoming proteins via hydrophobic interactions thereby mediating protein translocation across the outer membrane by a 'holding trap' rather than a 'folding trap' mechanism.

*For correspondence: hannes. herrmann@biologie.uni-kl.de

**Competing interests:** The authors declare that no competing interests exist.

## Introduction

In most cellular compartments, cysteine residues are predominantly present in the reduced state. In contrast, the bacterial periplasm, the endoplasmic reticulum (ER) and the IMS of mitochondria contain dedicated oxidation machineries (disulfide relays) to introduce disulfide bonds into a broad range of substrate proteins (*Riemer et al., 2009*; *Kadokura et al., 2003*; *Bulleid and Ellgaard, 2011*; *Stojanovski et al., 2012*; *Modjtahedi et al., 2016*; *Riemer et al., 2015*). At least in the case of the periplasm and the ER, disulfide bond formation presumably serves the function of stabilizing the structures of (secretory) proteins. While oxidative protein folding in the periplasm and the ER is well characterized, the details of the mitochondrial disulfide bond formation are still elusive.

Mitochondria consist of about 600 (yeast) to 1500 (humans) nuclear encoded proteins (*Pagliarini et al., 2008*; *Vögtle et al., 2009*). Following their synthesis on cytosolic ribosomes, these proteins are recognized by receptors on the mitochondrial surface and threaded through mitochondrial protein translocases (*Schulz et al., 2015*; *Harbauer et al., 2014*; *Endo et al., 2011*; *Chacinska et al., 2009*; *Neupert and Herrmann, 2007*). Proteins destined for the matrix of mitochondria contain presequences (or matrix-targeting signals) at their N termini that target these proteins through the protein-conducting channels of the translocase of the outer membrane (TOM complex) and the inner membrane (TIM23 complex). After import into the mitochondria the presequences are proteolytically removed by the matrix processing peptidase MPP (*von Heijne, 1986*; *Vögtle et al., 2009*).

**eLife digest** Human, yeast and other eukaryotic cells contain compartments called mitochondria that perform several vital tasks, including supplying the cell with energy. Each mitochondrion is surrounded by an inner and an outer membrane, which are separated by an intermembrane space that contains a host of molecules, including proteins.

Intermembrane space proteins are made in the cytosol before being transported into the intermembrane space through pores in the mitochondrion's outer membrane. Many of these proteins have the ability to form disulfide bonds within their structures, which help the proteins to fold and assemble correctly, but they only acquire these bonds once they have entered the intermembrane space.

An enzyme called Mia40 sits inside the intermembrane space and helps other proteins to fold correctly. This Mia40-induced folding had been suggested to help proteins to move into the intermembrane space.

Mia40 contains two important regions: one region acts as an enzyme and adds disulfide bonds to other proteins, and the other region binds to the intermembrane space proteins. Peleh et al. have now generated versions of Mia40 that lack one or the other of these regions in yeast cells, and then tested to see if these mutants could drive proteins across the outer membrane of mitochondria. The results show that it is the ability of Mia40 to bind proteins – and not its enzyme activity – that is essential for importing proteins into the intermembrane space.

As disulfide bond formation is not critical for importing proteins into the intermembrane space, future studies could test whether Mia40 also helps to transport proteins that cannot form disulfide bonds. Presumably, Mia40 has a much broader relevance for importing mitochondrial proteins than was previously thought.

The targeting of proteins into the IMS, the compartment between the outer and the inner membrane of mitochondria, is less well understood. Some IMS proteins also contain N-terminal signals in the form of bipartite presequences which consist of a matrix-targeting signal followed by a hydrophobic stop-transfer domain. These proteins are arrested at the inner membrane before their mature domains are released into the IMS by proteolytic cleavage (*Glick et al., 1992*; *Herlan et al., 2004*; *Rojo et al., 1998*). The cytochrome $b_2$ protein of *Saccharomyces cerevisiae* is the best studied example for this stop-transfer targeting (*Glick et al., 1992*; *Hartl et al., 1987*; *Gärtner et al., 1995*).

Most IMS proteins do not contain N-terminal targeting signals but instead contain patterns of cysteine residues within their mature sequence that serve as targeting signals (*Koehler, 2004*; *Sideris and Tokatlidis, 2007*; *Sideris et al., 2009*; *Milenkovic et al., 2009*). In most cases, these proteins contain two pairs of cysteine residues that are either spaced by three or nine amino acid residues, therefore referred to as 'twin $Cx_3C$' and 'twin $Cx_9C$' proteins, respectively. In yeast, five 'twin $Cx_3C$' (also called small Tim proteins) play a role as chaperones for carrier proteins in the IMS (*Curran et al., 2002b*; *2002a*; *Koehler et al., 1998*; *Sirrenberg et al., 1998*; *Luciano et al., 2001*; *Vial et al., 2002*), and 13 'twin $Cx_9C$' proteins contribute to the stabilization and assembly of inner membrane proteins (*Potting et al., 2010*; *Vögtle et al., 2012*; *Longen et al., 2009*; *Modjtahedi et al., 2016*; *Horn et al., 2010*; *Bode et al., 2015*). But recently also proteins with different disulfide configurations were discovered (*Okamoto et al., 2014*; *Wrobel et al., 2013*; *Varabyova et al., 2013*; *Hangen et al., 2015*; *Klöppel et al., 2011*; *Wrobel et al., 2016*). The import of these proteins into mitochondria relies on the mitochondrial disulfide relay (also called MIA pathway) which employs two conserved, essential proteins, Erv1 and Mia40. Erv1 is an FAD-binding sulfhydryl oxidase which can 'generate' disulfide bonds de novo thereby transferring electrons either directly to molecular oxygen or to cytochrome *c* of the respiratory chain (*Lee et al., 2000*; *Dabir et al., 2007*; *Ang and Lu, 2009*; *Tienson et al., 2009*; *Bien et al., 2010*; *Mesecke et al., 2005*; *Allen et al., 2005*; *Bihlmaier et al., 2007*; *Fass, 2008*). The oxidoreductase Mia40 contains a redox-active cysteine-proline-cysteine (CPC) motif (*Terziyska et al., 2009*; *Chacinska et al., 2004*; *Banci et al., 2009*; *2010*; *Kawano et al., 2009*). Erv1 maintains this motif in an oxidized conformation which then permits the transfer of disulfide bonds to Mia40 substrates

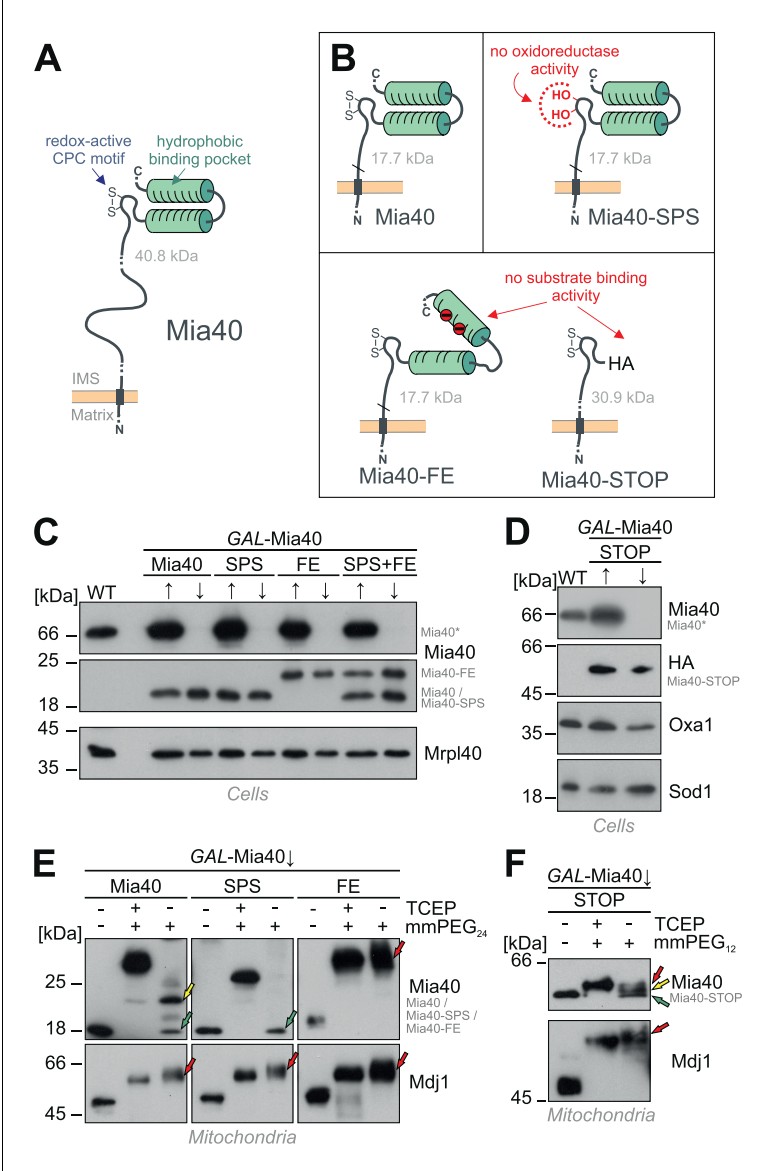

**Figure 1.** Generation of Mia40 mutants lacking either the oxidoreductase or substrate-binding activity. (**A**) Schematic representation of the Mia40 structure. (**B**) Structure of the Mia40 variants used for complementation studies. Molecular masses of the matured (MPP-cleaved) variants are indicated. (**C**, **D**) The indicated variants were expressed in *GAL*-Mia40 strains. Cells were grown in synthetic lactic acid-based medium containing 0.5% galactose (↑) or glucose (↓) to induce or repress the *GAL* promoter, respectively. The expression levels of the indicated Mia40 variants were assessed by Western blotting. Due to the increased net charge, Mia40-FE shows a reduced migration on SDS-PAGE. Mia40* depicts the endogenous full-length Mia40. (**E**) To monitor the redox state of the Mia40 variants, proteins of the indicated strains were TCA-precipitated, denatured in SDS, treated with the reducing agent TCEP and the alkylating compound mmPEG$_{24}$ and visualized by SDS-PAGE and Western blotting. Mdj1 is a matrix chaperone with 10 reduced cysteine residues which was used for control. Green arrows depict fully oxidized Mia40 species, yellow arrows the wild type Mia40 with the CPC reduced and the twin Cx$_9$C structure oxidized, and the red arrow a Mia40 protein in which the twin Cx$_9$C structure does not contain disulfide bonds. (**F**) The redox state of the Mia40-STOP variant was analyzed as described for **E** with the exception that mmPEG$_{12}$ was used for alkylation. The arrowheads depict different redox states of the protein suggesting that Mia40-STOP is partially oxidized.

during their translocation into the IMS (*von der Malsburg et al., 2011*). Mia40 substrates are often small proteins that are unstructured in the reduced form but very stable as soon as they are oxidized (*Morgan et al., 2009*; *Curran et al., 2004*; *2002a*; *Baker et al., 2012*; *Banci et al., 2009*; *2012*). It was proposed that oxidation traps the incoming polypeptides in the IMS so that oxidation-induced folding drives their net translocation into the IMS ('folding trap model') (*Lutz et al., 2003*;

*Koehler, 2004*). Mechanistically, this would be very different from protein translocation into the periplasm and the ER where the translocation of proteins is driven in ATP-dependent reactions by SecA or BiP, respectively, prior to and independent of their oxidative folding (*Matlack et al., 1999*; *Economou and Wickner, 1994*; *Taufik et al., 2013*). Moreover, the formation of mixed disulfides of Mia40 with incoming polypeptides was suggested to serve as a crucial reaction in the translocation reaction (*von der Malsburg et al., 2011*; *Longen et al., 2014*; *Bien et al., 2010*) that is critical to avoid the mistargeting of reduced IMS proteins to the cytosol (*Bragoszewski et al., 2015*; *Wrobel et al., 2015*).

Structural analysis revealed the presence of a hydrophobic substrate-binding pocket on the surface of Mia40 (*Figure 1A*); like the CPC motif this substrate-binding region of Mia40 is essential for its function in the import and folding of IMS proteins and mutants were shown to be inviable (*Banci et al., 2009*, *2010*; *Kawano et al., 2009*; *Weckbecker et al., 2012*). This region is essential for the binding and hence the oxidation of Mia40 substrates as well as for binding to Erv1. Mia40 substrates contain internal signals, known as MISS or ITS sequences, which specifically dock onto this binding region thereby selecting cysteine residues for interaction with the redox-active cysteine pair in Mia40 (*Koch and Schmid, 2014b*; *Sideris and Tokatlidis, 2007*; *Sideris et al., 2009*; *Peleh et al., 2014*; *Milenkovic et al., 2009*).

In this study, we analyzed the relevance of both structural elements of Mia40 in more detail. To this end, we generated Mia40 mutants which lack either the redox-active cysteine pair (Mia40-SPS) or the substrate-binding pocket (Mia40-FE and Mia40-STOP). Interestingly, the Mia40-SPS mutant still mediates protein import of Mia40 substrates with high efficiency. It also allows the accumulation of Mia40 substrates in mitochondria, albeit at reduced levels. Our observations suggest that trapping activity of Mia40 is essential since Mia40 serves as a 'trans-site receptor' whose hydrophobic binding to incoming polypeptides drives protein import into the mitochondrial IMS. Hence, protein oxidation in the IMS is not directly linked to translocation, and mechanistically is independent of the transport reaction, similar to the situation in the ER and the periplasm.

## Results

### The oxidoreductase and the holdase activity of Mia40 can be separated genetically

Studies on the structure of Mia40 (*Kawano et al., 2009*; *Banci et al., 2009*; *2010*) revealed the presence of two conserved functional elements, an N-terminal redox-active CPC motif and a C-terminal hydrophobic substrate-binding pocket (*Figure 1A*). We constructed Mia40 variants in which either the CPC motif was mutated to a redox-inactive SPS motif (*Figure 1B*, Mia40-SPS) or the hydrophobic binding region was compromised by replacement of two conserved phenylalanine residues at positions 315 and 318 by glutamate residues (*Figure 1B*, Mia40-FE). In order to distinguish these versions from the endogenous Mia40, we deleted the functionally irrelevant residues 211 to 283 from the membrane anchor in Mia40-SPS and Mia40-FE (*Terziyska et al., 2009*). Moreover, we constructed a Mia40 version in which the entire substrate-binding pocket (i.e. the twin $Cx_9C$ domain) was replaced by a hemagglutinin tag (*Figure 1B*, Mia40-STOP). All variants were expressed from single copy plasmids under control of the endogenous *MIA40* promoter in a *GAL*-Mia40 mutant in which the chromosomal *MIA40* gene is under control of a regulatable *GAL10* promoter (*Mesecke et al., 2005*). Growth of these strains on galactose or glucose caused the induction or repression, respectively, of chromosomal *MIA40* but did not affect the levels of the Mia40 variants expressed from the plasmids (*Figure 1C,D*).

Next we tested whether the Mia40 mutants still contain the two structural disulfide bonds, which are crucial for the functionality of the substrate-binding pocket (*Terziyska et al., 2009*). We isolated mitochondria from the different mutants after depletion of the endogenous Mia40 (*GAL*-Mia40↓), precipitated proteins with trichloroacetic acid (TCA) to denature them and to preserve their redox state, and incubated them with mmPEG$_{24}$ (*Figure 1E*) or mmPEG$_{12}$ (*Figure 1F*). These maleimide-based alkylating agents lead to mass shifts of about 1.2 and 0.7 kDa per alkylated thiol group, respectively. For Mia40 two species were observed both containing the two structural disulfides of the substrate-binding domain but differing in the redox state of the CPC motif (*Figure 1E*, left panel: green arrow, CPC oxidized; yellow arrow, CPC reduced). Whereas the Mia40-SPS mutant

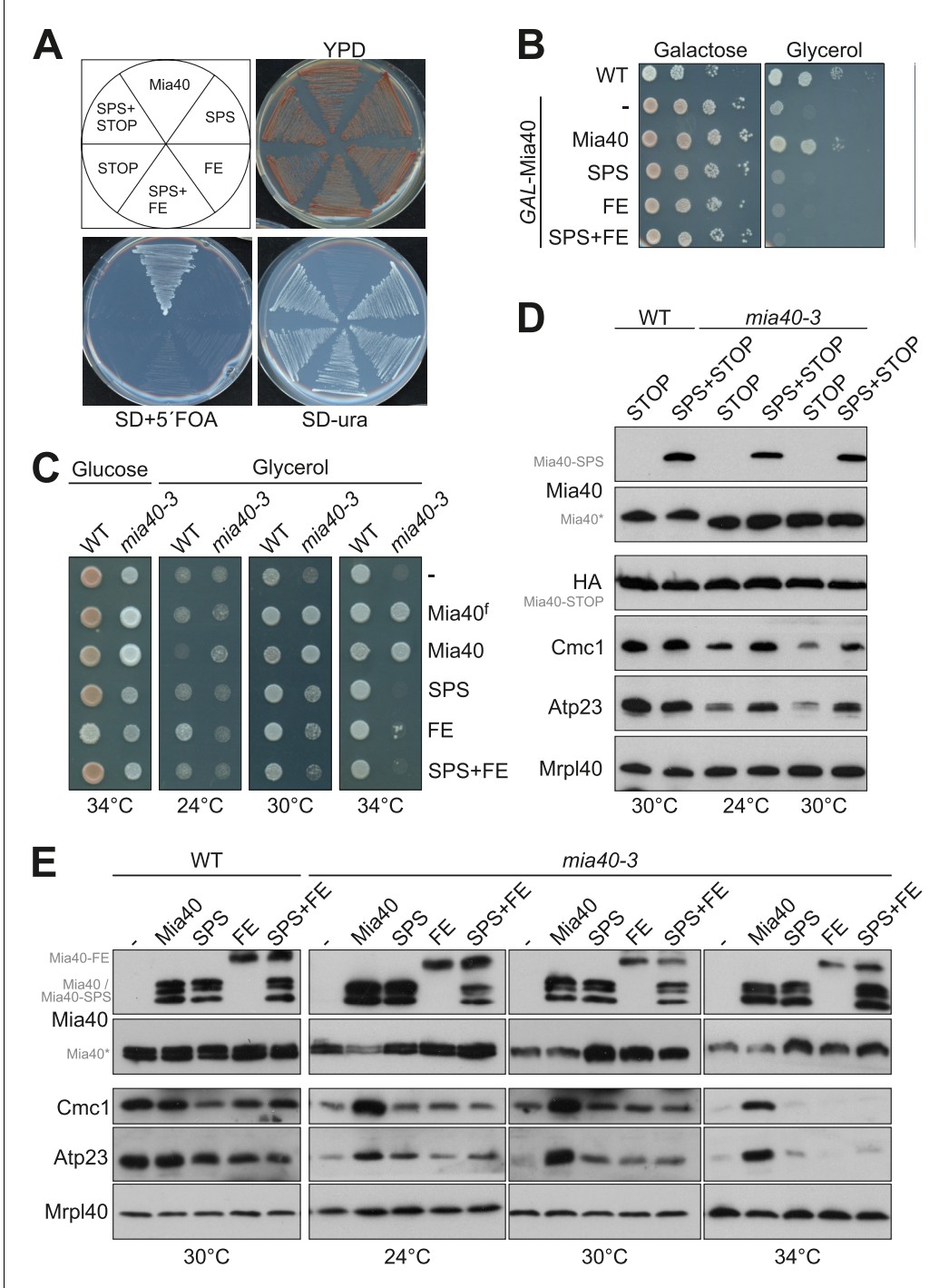

**Figure 2.** Mia40 mutants lacking either the oxidoreductase or the substrate-binding activity cannot cross-complement each other. (A) Δ*mia40* cells containing *MIA40* on a *URA3* plasmid were transformed with plasmids for expression of the indicated Mia40 variants. After growth on uracil-containing medium, loss of the Mia40-containing *URA3* plasmid was tested by growth on 5-fluoroorotic acid (5′FOA, which is converted to the toxic 5-fluorouracil in the presence of Ura3). Loss of the *URA3* plasmid was lethal except upon expression of wild type Mia40, indicating that none of the mutated Mia40 variants rescued the Δ*mia40* mutants, even if expressed in combinations. (B) Expression of wild type Mia40 but not of the Mia40 variants rescued the inability of *GAL*-Mia40 mutants to grow on glycerol medium. (C) Wild type cells and the temperature-sensitive *mia40-3* mutant were transformed with plasmids expressing full-length Mia40 (Mia40[f]) (*cf.* **Figure 1A**) or the indicated Mia40 variants (*cf.* **Figure 1B**). The mutated Mia40 variants did not allow growth at restrictive temperature. (D, E) Protein levels were analyzed by Western blotting showing that the different Mia40 variants were well expressed in the *mia40-3* mutant. Please note that only the expression of wild type Mia40 resulted in the efficient accumulation of Mia40 substrates such as Cmc1 and Atp23. However, small amounts of Cmc1 and Atp23 were also observed in the Mia40-SPS but not in the Mia40-FE or the Mia40-STOP cells.

contained a properly oxidized substrate-binding domain (*Figure 1E*, middle panel: green arrow), the structural disulfides were not formed in the Mia40-FE mutant (*Figure 1E*, right panel: red arrow) indicating that the negative charges prevented the folding of the substrate-binding domain. In the Mia40-STOP variant, different redox states of the CPC motif were observed (*Figure 1F*) indicating that its cysteine residues are partially oxidized even in the complete absence of the substrate-binding domain (cf. *Figure 1B*). We conclude that the different Mia40 variants can be expressed in the IMS of mitochondria and either lack the redox-active cysteine pair (but contain the correctly folded substrate-binding domain) or lack a functional substrate-binding domain (but contain the CPC motif).

## Mia40 variants lacking oxidoreductase or holdase activity cannot cross-complement each other

Next we tested if the Mia40 mutants can functionally replace the endogenous Mia40. To this end, we followed three different strategies. First, we followed a plasmid-shuffling approach in which we tested whether a Mia40-expressing *URA3* plasmid can be functionally replaced by plasmids that express the mutated Mia40 variants (*Figure 2A*). Second, we introduced the Mia40-expressing plasmids in a *GAL*-Mia40 strain and repressed the endogenous Mia40 (*Figure 2B*). And third, we tested complementation in a temperature-sensitive *mia40* mutant (*Chacinska et al., 2004*) at a restrictive temperature (*mia40-3*, *Figure 2C*). Although the different Mia40 variants were efficiently expressed in these strains and correctly targeted to the mitochondrial IMS (*Figure 2D,E* and data not shown), neither Mia40-SPS, nor Mia40-FE rescued the loss of the endogenous Mia40. The endogenous Mia40 could only be functionally replaced by an expression of wildtype Mia40.

It should be noted that Mia40 remained essential even upon simultaneous co-expression of Mia40-SPS and Mia40-FE or of Mia40-SPS and Mia40-STOP. Hence, Mia40 variants lacking either the redox-active cysteine pair or the substrate-binding pocket cannot cross-complement each other and obviously both elements need to be present in close proximity in the same protein.

## The chemical oxidant diamide can partially rescue the growth defect of Mia40-SPS

Cross-complementation of Mia40-SPS and Mia40-FE might have failed because the redox-active disulfide and the substrate-binding pocket are not in close proximity (at the necessary concentrations) required to allow efficient substrate oxidation. We therefore tested whether the addition of a chemical oxidizer can support the growth of the Mia40-SPS mutant. Diamide is a thiol-specific chemical oxidant that induces disulfide bond formation in proteins (*Frand and Kaiser, 1998*; *Hansen et al., 2009*) and which was shown to efficiently oxidize proteins of the mitochondrial IMS when added to the growth medium of yeast cells (*Kojer et al., 2012*; *2015*). We transformed *GAL*-Mia40 cells with plasmids for the individual or simultaneous expression of Mia40-SPS and Mia40-FE, grew the cells on glucose medium and placed a diamide-containing filter onto the plates (*Figure 3A*). Interestingly, we observed a ring of growing colonies around the diamide-containing filter in cells expressing Mia40-SPS but not in cells expressing Mia40-FE. We conclude that diamide is able to partially rescue the growth defect of the oxidation-deficient Mia40 mutant. A similar growth stimulation by diamide was also observed in liquid cultures (*Figure 3B*). When *GAL*-Mia40 cells are grown in lactic acid-based medium containing 0.5% glucose they are unable to proliferate until the glucose concentration is strongly reduced by metabolism which takes about 24 hr under the conditions we used (*Figure 3B*, left panels). However, upon expression of the Mia40 variant the cells start growing already after a short lag phase (*Figure 3B*, middle panels). Short lag phases were likewise observed upon expression of Mia40-SPS, however, only in the presence of sublethal concentrations of diamide (*Figure 3B*, right panels). Thus, in the presence of a chemical oxidizer, the hydrophobic binding pocket of Mia40 is apparently sufficient – and still essential - for viability.

Why does diamide suppress the growth defect of Mia40-SPS mutants? Either it supports the functionality of the Mia40-SPS protein (e.g., by the formation of the structural disulfides in the twin $Cx_9C$ domain of this protein), or it might act downstream in oxidizing proteins that were initially imported by Mia40-SPS. To test whether diamide directly supports the functionality of the Mia40-SPS protein, we isolated mitochondria containing either Mia40-SPS or Mia40-FE, pretreated them with increasing amounts of diamide and then added a radiolabeled cysteine-less model substrate of Mia40, 10CS.

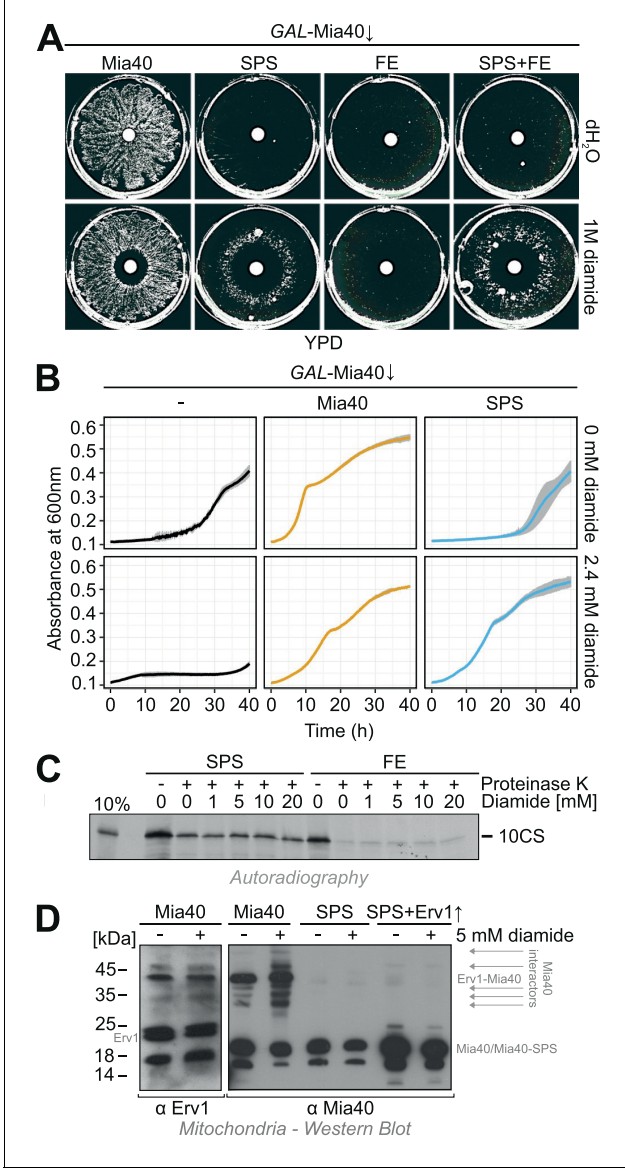

**Figure 3.** Diamide partially rescues the growth defect of the Mia40-SPS mutant. (**A**) The indicated strains were grown in synthetic lactic acid-based medium containing 0.5% glucose for three days to deplete endogenous Mia40 and spread on glucose plates. 10 µl water or 1 M diamide was applied onto a filter dish in the middle of the plate. Note the ring-like growth of the Mia40-SPS mutant around diamide-containing filter. (**B**) *GAL*-Mia40 mutants lacking or containing plasmids for expression of the Mia40 or the Mia40-SPS variant were grown for 72 hr in synthetic lactic acid-based medium supplemented with 0.5% glucose and diluted to OD 0.1 in the same medium lacking or containing 2.4 mM diamide. Growth at 30°C under constant shaking was then analyzed continuously in a multiwell absorption reader (BioTek ELx808, BMG Labtech). Error bars represent SD with n = 4. (**C**) Mia40-SPS- and Mia40-FE-containing mitochondria were incubated in import buffer for 5 min in the presence of the indicated concentrations of diamide. Then the radiolabeled cysteine-less Atp23 variant 10CS was added. After incubation for 15 min at 25°C, non-imported protein was removed by protease treatment. Mitochondria were washed and subjected to SDS-PAGE and autoradiography. (**D**) Mitochondria were isolated from glucose-grown *GAL*-Mia40 cells expressing the indicated Mia40 variants. They were treated with or without 5 mM diamide for 10 min at 30°C, reisolated, resuspended in non-reducing sample buffer and analyzed by Western blotting. Many disulfide-linked adducts are observed with wild type Mia40, particularly after incubation with diamide, but not with Mia40-SPS, not even when Erv1 is overexpressed.

This protein represents a mutant version of the IMS protease Atp23 in which all ten cysteine residues were replaced by serine residues (*Weckbecker et al., 2012*). The import of this protein into the mitochondria was assessed by analysis of the protease-protected 10CS fraction after 15 min of incubation (*Figure 3C*). Interestingly, Mia40-SPS was able to efficiently promote the import of this protein whereas 10CS was not taken up by Mia40-FE mitochondria. However, diamide did not affect

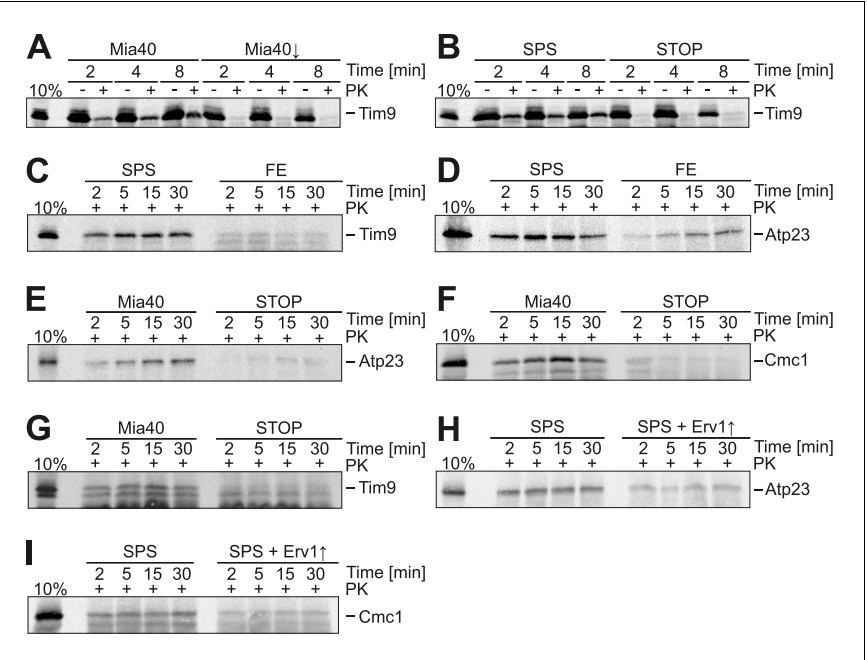

**Figure 4.** Substrate-binding by Mia40 is necessary and sufficient for protein translocation into the IMS. (**A**) Depletion of Mia40 blocks import of Tim9. Radiolabeled Tim9 was incubated at 25°C for the times indicated with mitochondria isolated from Mia40-depleted cells lacking or containing a plasmid for expression of the Mia40 variant. After treatment with proteinase K (PK) mitochondria were reisolated, washed and subjected to SDS-PAGE and autoradiography. 10% of the radiolabeled protein used per import sample is shown for comparison. (**B–I**) In vitro import reactions with radiolabeled Tim9, Atp23 and Cmc1 into the mitochondria of the indicated strains.

the import into either mutant suggesting that it has no effect on the functionality of Mia40-SPS or Mia40-FE.

Western blotting of extracts of diamide-treated mitochondria suggests that diamide increases the number of disulfide-linked adducts on Mia40, but not on Mia40-SPS, even in the presence of overexpressed Erv1 (*Figure 3D*). Hence, diamide does not induce non-canonical disulfides between Mia40-SPS and imported proteins but presumably acts on the imported substrates after their translocation into the mitochondria thereby stabilizing their structure. In summary, we conclude that the redox-inactive Mia40-SPS mutant is still functional in the import of a cysteine-free model protein and sufficient for cell survival in the presence of the chemical oxidizer diamide.

## The redox-active cysteine pair of Mia40 is dispensable for protein import of Mia40 substrates

The observation that Mia40-SPS efficiently drives the import reaction of the 10CS protein inspired us to test whether this variant likewise supports the import of cysteine-containing Mia40 substrates. To this end, we imported radiolabeled Tim9 and Atp23, two proteins that are imported in a strictly Mia40-dependent reaction (*Chacinska et al., 2004*; *Naoé et al., 2004*; *Weckbecker et al., 2012*). Mia40-SPS was fully sufficient to drive the import reaction of both proteins (*Figure 4A–D*). We conclude that Mia40-driven oxidation is not critical for protein translocation into mitochondria. This was very surprising since the oxidative folding by Mia40 was proposed to drive the import of its substrates across the outer membrane ('folding trap model') (*Lutz et al., 2003*; *Koehler, 2004*; *Bihlmaier et al., 2007*). Whereas the redox-active cysteines were dispensable for protein import, the hydrophobic binding domain was essential since no import of Mia40 substrates was observed with the Mia40-STOP mutant (*Figure 4B, E–G*).

It is conceivable that the redox-active cysteine pair in Mia40 is not crucial because Erv1 might directly oxidize these proteins in the Mia40-SPS mutant. Indeed, it was reported that overexpression of Erv1 partially suppresses the growth defect of some Mia40 mutants (*Kawano et al., 2009*). However, when we overexpressed Erv1 in the Mia40-SPS background, we did not observe a stimulation,

but rather a considerable reduction of the import of Atp23 and Cmc1 into the IMS (*Figure 4H, I*). This reduction is most likely the consequence of a competitive binding of Erv1 and the incoming polypeptides to the substrate-binding pocket of Mia40 (*Banci et al., 2011*).

When Mia40-SPS was isolated by immunoprecipitation during the import reactions, considerable fractions of Tim9 and Cmc1 were co-isolated (*Figure 5A,B*), albeit not covalently bound via disulfides (*Figure 5C,D*). This indicates a tight association of the incoming proteins with the substrate-binding pocket of Mia40. In contrast, Mia40 substrates (Cox19, Tim9, Atp23) were not co-isolated with the Mia40-STOP variant (*Figure 5—figure supplements 1* and *2*). Tim9 and Cmc1 were also not coisolated with Erv1, not even when Erv1 was overexpressed in the Mia40-SPS background (*Figure 5—figure supplement 3*), indicating that Erv1 does not interact with these proteins directly or the interaction is too transient or weak to be detected. In summary, we conclude that the substrate-binding pocket of Mia40 is necessary and sufficient for protein import of Mia40 substrates whereas the redox activity of Mia40 is apparently dispensable.

## Mia40-SPS is sufficient for the accumulation of Mia40 substrates in mitochondria

Next, we tested whether Mia40-SPS can also mediate protein targeting to the IMS in vivo. The depletion of Mia40 from mitochondria results in the concomitant depletion of its substrates as shown in many previous studies (*Chacinska et al., 2004*; *Naoé et al., 2004*; *Mesecke et al., 2005*; *Allen et al., 2005*). However, when Mia40-SPS was expressed in Mia40-depleted cells, we observed that the levels of Mia40 substrates were partially restored (*Figure 6A*). When we tested the redox states of the Mia40 substrate Tim10 in the Mia40-SPS mutant we found, that both disulfides in Tim10 were properly formed (*Figure 6B,C*). This suggests that the redox-active CPC motif in Mia40 accelerates substrate oxidation, but it is not absolutely essential for protein oxidation in the IMS.

The presence of a functional substrate-binding pocket in Mia40 was critical for the accumulation of Mia40 substrates in vivo as Atp23 and Cmc1 were not detectable in Mia40-STOP mitochondria unless Mia40-SPS was co-expressed (*Figure 6D*).

The ability of Mia40-SPS to partially restore the levels of Cox19, Atp23, and Cmc1 was also observed in cells of temperature-sensitive *mia40* mutant that were grown under restrictive conditions (*Figure 6E*), again indicating that the substrate-binding pocket is sufficient for protein import. Overexpression of Erv1 did not further increase the levels of IMS proteins in the Mia40-SPS mutant but rather strongly diminished them (*Figure 6F*). This again points to a competitive binding of Erv1 and Mia40 substrates to the crucial substrate-binding domain of Mia40-SPS.

Next, we followed the oxidation of newly synthesized Cox19-HA in a pulse chase experiment in wild type and *mia40-4* cells (*Figure 6G,H*). To this end, translation products were radiolabeled in yeast cells for 3 min. Then, the labeling was stopped by washing the cells and by the addition of an excess of non-radioactive methionine. After different times of incubation, samples were taken and subjected to alkylation with mmPEG$_{24}$. Then, Cox19-HA was immunoprecipitated and analyzed by SDS-PAGE. In the wild type, Cox19-HA was rapidly oxidized and within less than 2 min the two disulfide bonds were formed. In the temperature-sensitive *mia40-4* mutant, the oxidation of Cox19-HA was considerably slower and only after 4 to 8 min, half of the protein was oxidized. The oxidation was not accelerated by expression of Mia40-SPS. This confirms that the positive effect on the import and the accumulation of IMS proteins in the Mia40-SPS mutant is independent of their oxidation.

From the partial restoration of the IMS import by Mia40-SPS we conclude that the primary function of Mia40 is a role as a trans-site receptor that drives the translocation of proteins across the outer membrane. In addition, it also facilitates protein oxidation, however, this function appears to be important only after the translocation reaction, presumably to fold IMS proteins into a functional and protease-resistant confirmation.

## The level of Mia40 is limiting for the import of IMS proteins

The observation that the depletion of Mia40 leads to a rapid concomitant depletion of Mia40 substrates inspired us to test whether – on the contrary - overexpression of Mia40 increases the levels of Mia40 substrates. As shown in *Figure 7A*, overexpression of Mia40 leads indeed to much higher cellular levels of Mia40 substrates, such as Atp23, Tim10 and Cmc1. Thus, the endogenous levels of Mia40 are obviously rate-limiting under physiological conditions, suggesting that only a fraction of

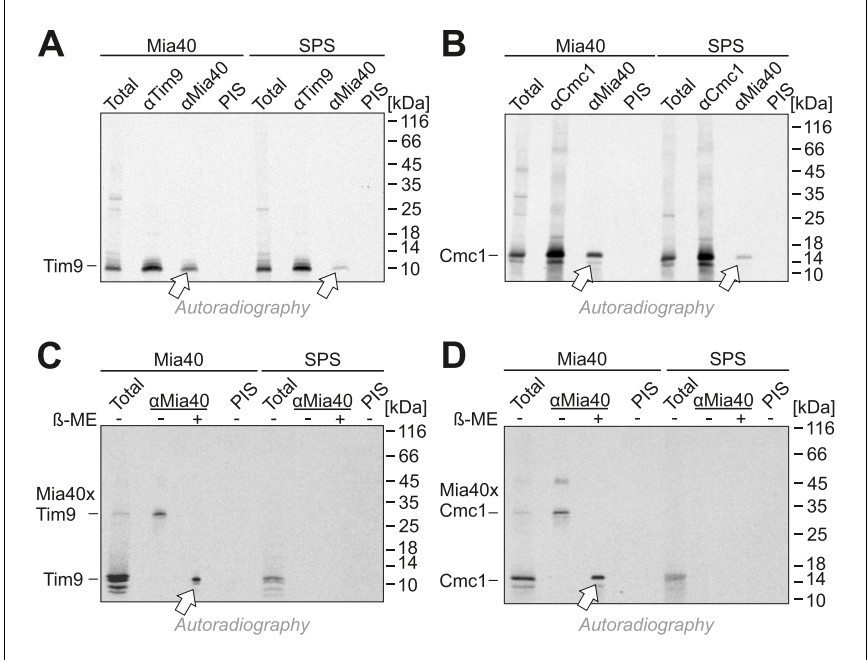

**Figure 5.** Mia40-SPS binds to import intermediates of Tim9 and Cmc1. (**A, B**) Radiolabeled Tim9 and Cmc1 were incubated for 2 min with isolated mitochondria containing Mia40 or Mia40-SPS. Mitochondria were reisolated and lysed with 1% SDS. The extract was used for immunoprecipitation with Tim9-, Cmc1- or Mia40-specific antibodies or with preimmune serum (PIS). Disulfide bonds were reduced with DTT. Radioactive proteins were visualized by SDS-PAGE and autoradiography. Total samples contain 10% of the material used per immunoprecipitation reaction. Arrows depict radiolabeled proteins pulled down with Mia40 and Mia40-SPS. (**C, D**) Mia40- or Mia40-SPS-containing mitochondria were treated for 10 min with 5 mM DTT at 30°C, tenfold diluted and incubated with radiolabeled Tim9 or Cmc1 for 2 min. Reisolated mitochondria were lysed with 1% SDS before the extract was used for immunoprecipitation using Mia40-specific antibodies or preimmune serum for control. Reducing (+ ß-mercaptoethanol, ß-ME) or non-reducing samples were analyzed by SDS-PAGE and autoradiography. See figure supplements for additional panels.

The following figure supplements are available for figure 5:

**Figure supplement 1.** The substrate-binding cleft but not the redox-active CPC motif of Mia40 is sufficient for the import of Tim9 and Cox19.

**Figure supplement 2.** Cysteines in the CPC motif of Mia40 are dispensable for the import of Tim9 and Atp23.

**Figure supplement 3.** Tim9 and Cmc1 form mixed disulfides with Mia40 but not with Mia40-SPS or Mia40-STOP.

the Mia40 substrates that are initially synthesized in the cytosol finally accumulate as stable proteins in vivo and that overexpression of Mia40 increases this fraction. This was also obvious when isolated mitochondria of these strains were analyzed (*Figure 7B*). We even observed increased amounts of cytochrome $b_2$, a protein that is imported into the IMS on a stop-transfer pathway (*Glick et al., 1992*; *Gärtner et al., 1995*; *Esaki et al., 1999*) and that was not expected to be influenced by the presence of Mia40. However, it was previously proposed that in *Candida albicans*, the homolog of cytochrome $b_2$ might be imported in a Mia40-driven reaction (*Hewitt et al., 2012*).

We next tested the import efficiencies of different Mia40 substrates with wild type and Mia40-upregulated mitochondria (*Figure 7C*). We found consistently, that the upregulation of Mia40 considerably improved the import of Cmc1, Atp23 and Tim9. This confirms a rate-limiting function of Mia40 during the import of IMS proteins.

In contrast, upregulation of Mia40 did not increase, but rather reduced, the import of cytochrome $b_2$ (*Figure 7D*), a well-characterized substrate of the presequence pathway (*Hartl et al., 1986*). Depletion of Mia40 did hardly affect the import of cytochrome $b_2$, however, it impaired its processing. This is presumably explained by the fact that the Mia40 substrate Som1 is a component of the inner membrane protease complex that mediates the maturation of cytochrome $b_2$ (*Jan et al., 2000*; *Nunnari et al., 1993*). Immunoprecipitation experiments did not indicate a direct binding of

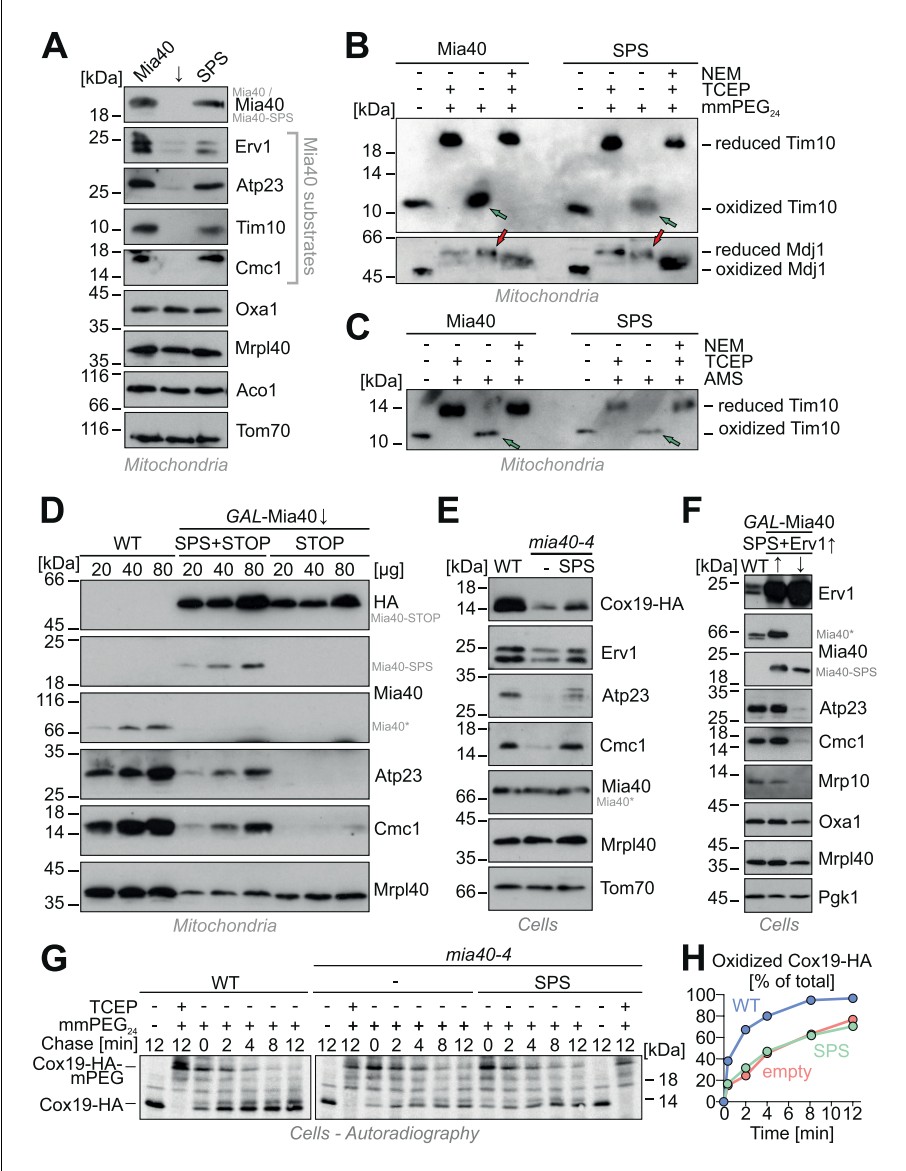

**Figure 6.** Expression of Mia40-SPS leads to the accumulation of Mia40 substrates in mitochondria. (**A**) Mitochondria were isolated from Mia40-depleted cells (↓) which expressed Mia40 or Mia40-SPS. The steady state levels of Mia40 substrates and control proteins were analyzed by Western blotting. (**B**, **C**) To determine the redox state of Tim10, proteins of Mia40- and Mia40-SPS-containing mitochondria were TCA-precipitated, denatured in SDS, treated with or without the reducing agent TCEP and the alkylating compounds mmPEG$_{24}$ (**B**) or AMS (**C**) and visualized by SDS-PAGE and Western blotting. An inverse shift was achieved by blocking reduced thiols with N-ethylmaleimide (NEM) prior to TCA precipitation. Green arrows depict fully oxidized Tim10, red arrows the matrix chaperone Mdj1 which does not contain disulfide bonds. (**D**) Western blots of mitochondrial extracts of the indicated strains. Please note that expression of Mia40-SPS but not of Mia40-STOP leads to accumulation of low levels of Atp23 and Cmc1 in mitochondria. (**E**) Wild type and *mia40-4* cells were transformed with a Cox19-HA-expressing plasmid (***Bode et al., 2015***) allowing detection of Cox19 with hemagglutinin (HA) antibodies in whole cell extracts. Levels of Mia40 substrates and other mitochondrial proteins were analyzed by Western blotting in the indicated cells after growth at 34°C. (**F**) Western blot analysis of protein levels in wild type and *GAL*-Mia40 cells expressing Mia40-SPS variant and high levels of Erv1. Arrows up and down indicate the expression or depletion of Mia40 in this strain, respectively. (**G**, **H**) To follow the oxidation of Cox19 in vivo, cells of the indicated strains were pulse-labeled for 3 min with [$^{35}$S]-methionine and chased with cold methionine for different times (***Kojer et al., 2015***). TCA-precipitated protein extracts were treated with mmPEG$_{24}$, immunoprecipitated with hemagglutinin-antibodies and analyzed by SDS-PAGE and autoradiography.

Mia40 to import intermediates of cytochrome $b_2$ (not shown) and we assume that the up- and down-regulation of Mia40 regulates the levels of cytochrome $b_2$ indirectly. Thus, the strong rate-limiting function of Mia40 on the import of its substrates is obviously highly relevant to the protein

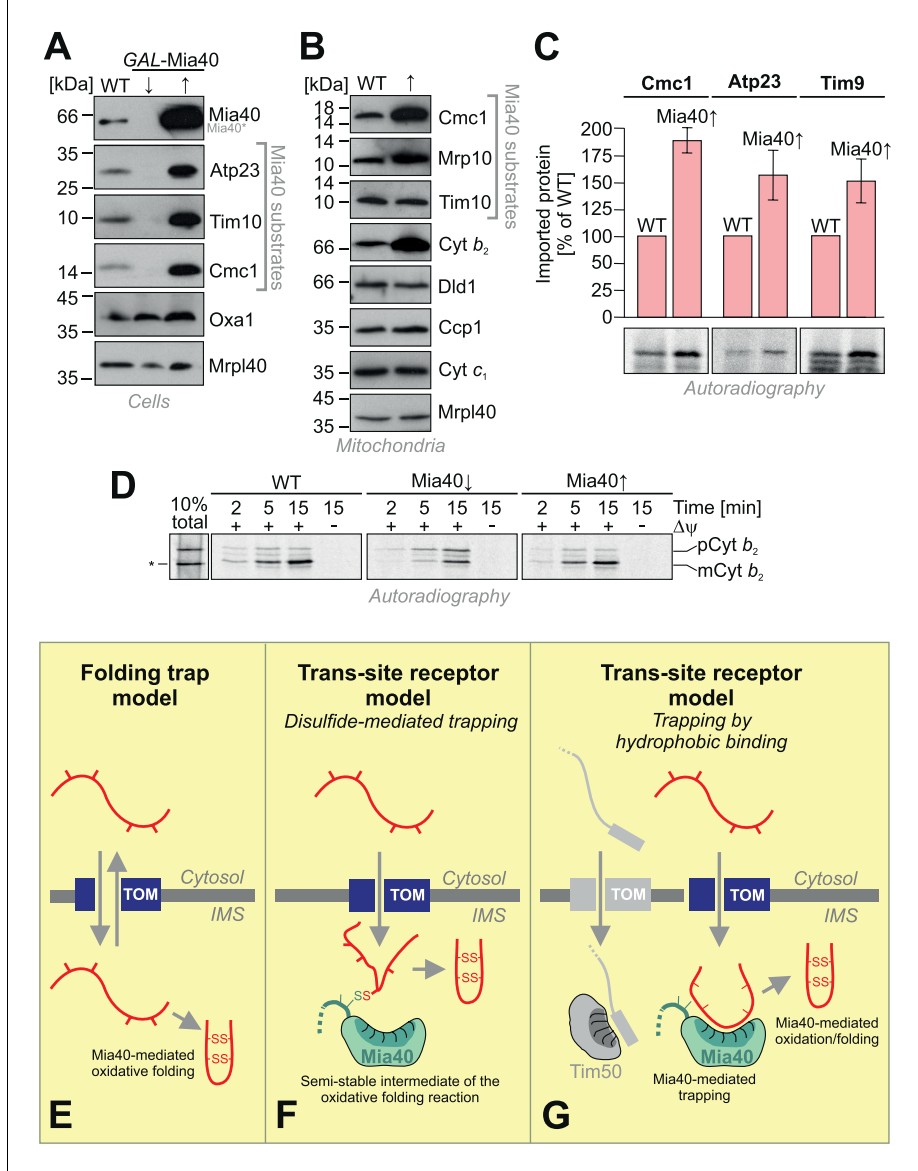

**Figure 7.** Mia40 is rate-limiting for the import of IMS proteins. (**A**) Western blot analysis of mitochondrial protein levels in galactose-grown wild type (WT) cells and in *GAL*-Mia40 cells grown in lactic acid-based media containing 0.5% glucose (↓) or galactose (↑). Overexpression of Mia40 leads to strongly increased steady-state levels of its substrates such as Atp23, Tim10 and Cmc1. (**B**) Mitochondria were isolated from WT and *GAL*-Mia40 cells cultured as described in **A**. Steady state levels of IMS proteins were analyzed by Western blotting. Please note the increased level of cytochrome $b_2$ upon overexpression of Mia40. (**C**) Quantifications of the amounts of imported radiolabeled proteins from in vitro import experiments with the Mia40 substrates Cmc1, Atp23 and Tim9. Error bars correspond to SEM with n = 4. Radioactive signals of one representative experiment are shown. (**D**) In vitro import reaction with radiolabeled cytochrome $b_2$ into mitochondria of the indicated strains. A shorter ('pseudomature') form of cytochrome $b_2$ that is formed by initiation at the second ATG codon is indicated by an asterisk. (**E–G**) Schematic representations of the 'folding trap model' and the 'trans-site receptor' models. According to the folding trap model, protein translocation into the IMS is driven by the formation of disulfides which prevents back-translocation into the cytosol. The 'trans-site receptor model' suggests that translocation is driven by the affinity of incoming proteins to the substrate-binding pocket of Mia40. It was suggested that the disulfide-mediated trapping is essential for the translocation reaction. However, our results shown here suggest that the trapping via hydrophobic binding of the incoming polypeptide to Mia40 is essential and sufficient for the translocation reaction. Oxidative folding is only a subsequent reaction which can facilitate the release from Mia40 and increase the proteolytic stability of imported proteins. See discussion for details.

composition of the IMS in general and critical even for proteins which are imported in a Mia40-independent fashion.

## Discussion

Mia40 is a highly conserved IMS protein which consists of two functionally and structurally distinct elements. Here we show that both the redox-active cysteine pair and the substrate-binding region of Mia40 are essential and mutants lacking individual elements cannot cross-complement each other. However, we found that cysteines in the CPC motif of Mia40 are dispensable for the import reaction of all Mia40 substrates we tested. This was unexpected because it was suggested that the mitochondrial disulfide relay operates as a folding trap (*Figure 7E*) (*Lutz et al., 2003*; *Koehler, 2004*). This model postulated that substrates would enter the IMS via the TOM pore in a reversible reaction. The reversible nature of the outer membrane translocation of Mia40 substrates is experimentally well documented (*Lutz et al., 2003*; *Bragoszewski et al., 2015*). According to the folding trap hypothesis, the directivity of the import reaction is driven by the introduction of disulfide bonds thereby locking proteins in a folded confirmation which traps them in the IMS (*Mesecke et al., 2005*; *Bihlmaier et al., 2007*). Based on the identification of the MISS/ITS signal as specific recognition sites in Mia40 substrates (*Koch and Schmid, 2014b*; *Sideris and Tokatlidis, 2007*; *Sideris et al., 2009*; *Peleh et al., 2014*; *Milenkovic et al., 2009*) and the observation of Mia40-linked import intermediates that span the TOM complex (*von der Malsburg et al., 2011*) it was suggested that Mia40 serves as an IMS-located receptor protein critical for outer membrane translocation of its substrates (*Figure 7F*). According to both models, disulfide bond formation by Mia40 is the critical feature for its role in protein translocation. However, our observation that the Mia40-SPS variant can efficiently drive the import reaction suggests that the affinity to the substrate-binding pocket provides the driving force for the translocation reaction rather than protein oxidation. Hence, we propose that Mia40 serves as a 'trans-site receptor' on the IMS site of the TOM complex that mediates protein translocation across the outer membrane (*Figure 7G*). At present it is not known whether the trapping function of Mia40 is restricted to disulfide-containing proteins or plays a broader, more general role in mitochondrial protein import. However, the recent identification of non-canonical substrates of Mia40 (*Petrungaro et al., 2015*; *Okamoto et al., 2014*; *Wrobel et al., 2013*; *Hangen et al., 2015*) might indeed suggest a much more general role of Mia40 in protein translocation across the outer membrane than previously expected.

The oxidoreductase activity of Mia40 is obviously important, but it apparently plays a role after translocation. This is consistent with a recent study which showed that mutants of small Tim proteins, in which all cysteine residues were replaced by serine residues, accumulated to wild type levels in the IMS as soon as the iAAA protease Yme1 was deleted (*Baker et al., 2012*).

Surprisingly, the presence of diamide can partially suppress the growth defect of Mia40-SPS cells suggesting that the substrate-specificity of the mitochondrial disulfide relay is not absolutely necessary. The observation of oxidized Tim10 in the Mia40-SPS mutant indicates that the CPC of Mia40 is not absolutely essential for protein oxidation in the IMS. Whether Erv1 can directly oxidize IMS proteins or whether other factors might serve as oxidoreductases here is not known, but it will be interesting to test the relevance of the recently identified thioredoxins and glutaredoxins in the IMS in this process (*Vögtle et al., 2012*; *Kojer et al., 2015*). However, it should be noted that, even in the presence of diamide, Mia40-SPS cells were very sick and protein oxidation in the IMS is obviously still an essential process.

Our observations indicate that Mia40 is rate-limiting for the import into mitochondria as overexpression of Mia40 leads to a considerable increase of the levels of many Mia40 substrates. Apparently, a fraction of the Mia40 substrates that are initially synthesized in the cytosol are degraded in the cytosol or the IMS, an assumption that is supported by pulse labeling experiments in whole cells (*Wrobel et al., 2015*; *Longen et al., 2014*; *Kojer et al., 2015*; *Fischer et al., 2013*). Increased levels of Mia40 might reduce the degradation either by accelerating the import or the folding reaction. In contrast, up- or down-regulation of the levels of Erv1 had no positive effect on the steady state levels of IMS proteins again supporting the idea that the trapping rather than the oxidase function of Mia40 is decisive (*Bien et al., 2010*; *Mesecke et al., 2005*). The strong increase in Mia40 substrates in Mia40-overexpressing cells suggests that more Mia40 leads to more import sites for Mia40 substrates. It is conceivable that mitochondria contain specific import sites for different types of preproteins which differ at the level of the 'trans-site receptors' in the IMS (*Figure 7G*). Thus, matrix-targeted proteins might specifically encounter TOM complexes that are associated to Tim50 (*Lytovchenko et al., 2013*; *Marom et al., 2011*) whereas IMS proteins might use TOM complexes

that are in proximity to Mia40. This hypothesis is supported by studies on the Mic60 subunit of the MICOS complex which presumably tethers Mia40 to a subpopulation of the TOM complex (*von der Malsburg et al., 2011*; *Herrmann, 2011*; *Varabyova et al., 2013*). The existence of a discrete translocation route into the IMS was recently proposed for Mia40 substrates on the basis of *in organello* competition import experiments (*Gornicka et al., 2014*) and, even much earlier, for the most abundant IMS protein, cytochrome *c* (*Wiedemann et al., 2003*; *Diekert et al., 2001*; *Mayer et al., 1995*).

The bacterial periplasm, the ER and the mitochondrial IMS are the three compartments in which dedicated machineries mediate the oxidative folding of a broad range of proteins (*Riemer et al., 2009*). Members of the thioredoxin protein family mediate protein oxidation in the periplasm (DsbA) and the ER (PDI). Apparently, the thioredoxin-based oxidation system of the periplasm of the endosymbiont that served as the ancestor of mitochondria was replaced during evolution by the Mia40 system. This replacement during evolution was hardly driven to increase protein oxidation in the IMS since Mia40 is a poor catalyst: in its oxidizing power, it is weaker than DsbA, and in forming correct disulfides, it is slower than PDI (*Koch and Schmid, 2014b*; *2014a*; *Hudson and Thorpe, 2015*; *Tienson et al., 2009*). However, Mia40 binds much stronger and for much longer interaction times to its substrates than DsbA or PDIs (*Koch and Schmid, 2014b*; *Mesecke et al., 2005*; *Naoé et al., 2004*; *Chacinska et al., 2004*). It therefore appears likely that the ability of Mia40 to trap incoming preproteins efficiently at the trans-site of the TOM complex made Mia40 the superior oxidoreductase to mediate the translocation and oxidation of IMS proteins.

## Materials and methods

### Yeast strains and plasmids

All yeast strains used in this study were based on the wild type strain YPH499 (*Sikorski and Hieter, 1989*) including the regulatable *GAL*-Mia40 strain (*Mesecke et al., 2005* #5543) and the temperature-sensitive *mia40-3* and *mia40-4* strains (*Chacinska et al., 2004* #5355). To inactivate Mia40 in the temperature-sensitive *mia40* strains, the culture was shifted to 37°C for 17 hr. Yeast strains were either grown on synthetic media containing 2% galactose, in synthetic lactic acid-based medium (containing 2% lactic acid and 0.5% galactose or glucose) or in YP (1% yeast extract, 2% peptone) medium containing 2% galactose, glucose or glycerol (*Peleh et al., 2014* #7089).

For expression of the different Mia40 variants (Mia40, Mia40-SPS, and Mia40-FE) the *MIA40* promoter and sequences corresponding to the protein sequence of residues 1–70 and 284–403 were cloned into the single-copy plasmids pRS314, pRS315 or pRS316. Point mutations within *MIA40* were introduced by site-directed mutagenesis (*Weckbecker et al., 2012*). The resulting mutations were confirmed by sequencing. For expression of the Mia40-STOP variant the *MIA40* promoter and the sequence corresponding to amino acid residues 1–306 were cloned into pRS315.

### Alkylation shift experiments for redox state detection

Mitochondria were isolated as described (*Peleh et al., 2015*). To analyze the redox state of cysteine residues, mitochondrial proteins were TCA precipitated and treated as described (*Peleh et al., 2014*). For modification, final concentrations of 15 mM mmPEG$_{24}$, mmPEG$_{12}$ and AMS were used to modify reduced thiols. The pulse-chase labeling of Cox19 was performed as described (*Kojer et al., 2015*).

### Import of radiolabeled proteins into isolated mitochondria

The import reactions were performed as described (*Mesecke et al., 2005*) in the following import buffer: 1 M sorbitol, 100 mM Hepes pH 7.4, 160 mM KCl, 20 mM magnesium acetate, 4 mM KH$_2$PO$_4$. In addition 2 mM ATP, 2 mM NADH, 10 mM creatine phosphate, 100 µg/ml creatine kinase, 2 mM malate, and 2 mM succinate were supplied to energize the mitochondria. To keep the radiolabeled proteins in the reduced unfolded state 5 mM GSH and 5 mM EDTA were added to the import mix.

## Co-immunoprecipitation

To immunoprecipitate import intermediates with Mia40 or Erv1, import reactions were carried out for 2 min at 25°C. Mitochondria were reisolated by centrifugation (20,000 xg for 20 min at 4°C) and resuspended in lysis buffer I (30 mM Tris/HCl pH 8, 100 mM NaCl, 1% SDS, 2 mM PMSF). The extract was diluted tenfold in lysis buffer II (30 mM Tris/HCl pH 8, 100 mM NaCl, 1% Triton X-100, 2 mM PMSF). After incubation for 10 min at 4°C, the extract was cleared by a clarifying spin (20,000 xg for 10 min at 4°C). Antibodies against Mia40, Erv1, Atp23, Tim9, Cmc1, Cox19 or hemagglutinin were coupled to protein A-sepharose beads. The beads were incubated with the mitochondrial extract at 4°C, washed three times with the lysis buffer II (2,000 xg for 2 min at 4°C), resuspended in SDS-sample buffer and boiled for 5 min at 96°C. Samples were analyzed by SDS-PAGE and autoradiography.

## Acknowledgements

We thank Sabine Knaus and Nihan Ates for technical assistance, Bruce Morgan and Jan Riemer for discussion and Agnieszka Chacinska, Dejana Mokranjac and Klaus Pfanner for yeast strains, reagents and antibodies. This study was funded by grants of the Deutsche Forschungsgemeinschaft (He2803/4-2, SPP1710 and IRTG1830) and the BioComp initiative of Rheinland-Pfalz.

## Additional information

### Funding

| Funder | Grant reference number | Author |
| --- | --- | --- |
| Deutsche Forschungsgemeinschaft | He2803/4-2 | Johannes M Herrmann |
| Deutsche Forschungsgemeinschaft | IRTG1830 | Johannes M Herrmann |
| Deutsche Forschungsgemeinschaft | SPP1710 | Johannes M Herrmann |
| Research initiative of Rheinland-Pfalz | BioComp | Johannes M Herrmann |

The funders had no role in study design, data collection and interpretation, or the decision to submit the work for publication.

### Author contributions

VP, Conception and design, Acquisition of data, Analysis and interpretation of data, Drafting or revising the article; EC, JMH, Conception and design, Analysis and interpretation of data, Drafting or revising the article

### Author ORCIDs

Johannes M Herrmann, http://orcid.org/0000-0003-2081-4506

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
