## [Decision Letter]

Thank you for submitting your article "An essential trans-site receptor activity of Mia40 drives protein import into the intermembrane space of mitochondria" for consideration by *eLife*. Your article has been favorably evaluated by Vivek Malhotra (Senior editor) and three reviewers, one of whom, Nikolaus Pfanner, is a member of our Board of Reviewing Editors. The reviewers have opted to remain anonymous.

The reviewers have discussed the reviews with one another and the Reviewing Editor has drafted this decision to help you prepare a revised submission.

Summary:

In this manuscript Peleh et al. have analyzed the function of Mia40 in protein transport into mitochondria. Using a set of MIA40 mutant strains, they define how the proposed substrate binding domain and the redox active CPC motif promote protein translocation into the mitochondrial intermembrane space. They find that the redox active CPC motif is dispensable for the translocation process and that the substrate-binding domain can promote vectorial movement of the polypeptide independent. Thus, a predominant mechanism applied by Mia40 for substrate recognition is a trapping mode, for which disulfide bond formation is not necessary.

The results presented here provide new and unexpected mechanistic insight into the transport of intermembrane space proteins The surprising finding that transport is not linked to oxidative folding but rather facilitated by substrate binding to Mia40 's binding domain changes the current concept of the Mia40 transport pathway. The manuscript provides a new mechanistic model on the import driving forces. The findings are interesting not only for protein translocation specialists but also for a broader audience since they provide a conceptual framework to explain the involvement of Mia40 in the biogenesis of several non-redox dependent proteins.

The data are of high technical quality and the manuscript is very well written. The findings presented in this manuscript clearly change our view of intermembrane space-protein transport.

Essential revisions:

1) The point mutant FE that disrupts the hydrophobic substrate binding patch may not be redox-active since there is no oxidized protein detectable at steady state, in contrast to the STOP variant that lacks the entire substrate binding domain (Figure 1). The authors should either provide evidence that FE is redox-active or they should perform the import experiments shown in Figure 4 with the STOP mutant in order to assess any contribution of the CPC motif for translocation. While the experiment shown in Figure 5—figure supplement 1 indicates that the STOP mutant does not interact with substrate, an additional import experiment would be more sensitive and thus would be preferred. Also, the STOP mutant should be included in Figure 2.

2) The authors should discuss their data and the resulting novel conclusions for mitochondrial protein import more accurately. Statements such as "Mia40 serves as a trans-receptor" have been published in earlier papers. Based on earlier papers the critical point in initiating the oxidative folding pathway in mitochondria is the trapping by Mia40. The important novelty of the current work lies in the demonstration that trapping does not have to be mediated by a redox reaction, i.e. the formation of a disulfide bond, but can be facilitated by hydrophobic interactions of Mia40 with the incoming protein substrates. Several parts of the manuscript, including the title, abstract and model, should be rewritten to precisely specify these important conclusions.

3) It has been a long standing question in the redox field why Mia40-bound substrates could easily be detected under wild-type conditions whereas the substrate interactions of ER and bacterial oxidoreductases (PDI, DsbA) were highly transient. The identification of two different activities in Mia40 elegantly solves this problem. It will thus be good to discuss these general implications for the fields of protein translocation and oxidative protein folding in more detail, e.g. by addressing this open question (difference between Mia40 and PDI/DsbA) already in the Introduction.

4) Several papers have specified the cysteine residues of substrates in close proximity of the MISS/ITS signal for Mia40 recognition and showed the import deficiencies of substrates lacking this Cys residue. The in vivo accumulation of these proteins indicates, however, that this "recognition" defect may be bypassed by different means (either by use of other Cys residues or by folding that is not fully dependent on oxidation).

It would be good to have a WB to see the levels of Erv1 overexpression. Is it possible that substrate proteins form some other intermolecular disulfide bonds that do not involve the CPC motif of Mia40, i.e. with Erv1? Additionally, the possibility of substrate binding by Mia40-SPS via a disulfide bond should be tested in the presence of chemical reductant.

5) The authors did not provide a sufficient explanation concerning the action of diamide. One can assume that diamide should in fact improve the overall oxidative folding/mitochondrial accumulation of substrates; is this the case? Diamide may or may not improve the substrate binding to Mia40-wt or Mia40-SPS? The answer to this question would be important to specify the stage diamide works. Finally, does diamide influence Mia40 binding to Erv1 or substrate binding to Erv1? The mechanism for diamide should be proposed based on the experimental data.

6) The import experiments presented in Figure 4 should be described more systematically in the text. The graphs (Figure 4) look very convincing but the information on the number of experimental repetitions taken for the quantification is missing in the Figure Legend. The explanation given in a separate document is not sufficient. Quantifications and average from multiple experiments should be presented, including SEM optimally. Alternatively, it should be clearly stated that one experiment was quantified, and in this case the experiment should be presented in full, not only in the form of bars.

What is the purpose of Figure 4? On the one hand there is no WT control to compare with. On the other hand, the quantification of Tim9 import in SPS, FE and WT is depicted in Figure 4. If the authors want to show quantification of kinetics, they should include a WT or otherwise remove the panel.

The text does not mention the number of replicates used for the quantitations in Figure 4 and Figure 7 as well as for the growth curves in Figure 3 that appear to have error bars.

---

## [Author Response]

Essential revisions:

1) The point mutant FE that disrupts the hydrophobic substrate binding patch may not be redox-active since there is no oxidized protein detectable at steady state, in contrast to the STOP variant that lacks the entire substrate binding domain (Figure 1). The authors should either provide evidence that FE is redox-active or they should perform the import experiments shown in Figure 4 with the STOP mutant in order to assess any contribution of the CPC motif for translocation. While the experiment shown in Figure 5—figure supplement 1 indicates that the STOP mutant does not interact with substrate, an additional import experiment would be more sensitive and thus would be preferred. Also, the STOP mutant should be included in Figure 2.

Indeed, the redox state of the FE mutant is not clear. In the background of the four reduced cysteines in the Cx_9_C motif, our shift experiments do not allow to make clear conclusions on the redox state of the CPC motif. Therefore, we followed your suggestion and added novel experiments using the STOP mutant which is partially oxidized. First, we performed import experiments into Mia40-STOP mitochondria with Atp23, Cmc1 and Tim9. All three proteins were imported well in the presence of Mia40 but no import was observed into Mia40-STOP mitochondria. These novel experiments are shown in Figure 4.

Moreover, as requested we analyzed protein levels in mitochondria of the STOP mutant by Western blotting. In the novel Figure 2 we show that Cmc1 and Atp23 are reduced in the *mia40- 3* background in the presence of the STOP mutant whereas co-expression of the Mia40-SPS mutant largely restored the accumulation of these proteins. This supports our conclusion that the hydrophobic bind cleft is both essential and (to a certain degree) sufficient to import IMS proteins.

2) The authors should discuss their data and the resulting novel conclusions for mitochondrial protein import more accurately. Statements such as "Mia40 serves as a trans-receptor" have been published in earlier papers. Based on earlier papers the critical point in initiating the oxidative folding pathway in mitochondria is the trapping by Mia40. The important novelty of the current work lies in the demonstration that trapping does not have to be mediated by a redox reaction, i.e. the formation of a disulfide bond, but can be facilitated by hydrophobic interactions of Mia40 with the incoming protein substrates. Several parts of the manuscript, including the title, abstract and model, should be rewritten to precisely specify these important conclusions.

We agree with the referee in this point. A trapping function of Mia40 via disulfide bonds was proposed before and the specific recognition of MISS/ITS signals was of major importance to understand the role of Mia40 as a trans-site receptor. We now adapted the model in Figure 7 accordingly. Moreover, we made the title more specific (while trying to stay within the very short 120 character limit) and changed it to: “Mia40 is a trans-site receptor that drives protein import into the mitochondrial intermembrane space by hydrophobic substrate binding”. In addition, we changed the text throughout the manuscript to discuss previous studies in more detail. The hydrophobic-binding model and the disulfide-mediated trapping model (as shown by van der Malsburg et al.) are clearly not mutually exclusive.

3) It has been a long standing question in the redox field why Mia40-bound substrates could easily be detected under wild-type conditions whereas the substrate interactions of ER and bacterial oxidoreductases (PDI, DsbA) were highly transient. The identification of two different activities in Mia40 elegantly solves this problem. It will thus be good to discuss these general implications for the fields of protein translocation and oxidative protein folding in more detail, e.g. by addressing this open question (difference between Mia40 and PDI/DsbA) already in the Introduction.

We discuss this aspect in more detail in the Abstract and in the Discussion.

In the Abstract we added the sentence: “The oxidoreductases of the periplasm and the ER, DsbA and protein disulfide isomerase, are members of the thioredoxin superfamily which are structurally and phylogenetically distinct from the components of the mitochondrial disulfide relay.”

In the Discussion, the last paragraph reads: “The bacterial periplasm, the ER and the mitochondrial IMS are the three compartments in which dedicated machineries mediate the oxidative folding of a broad range of proteins (Riemer et al., 2009). […] It therefore appears likely that the ability of Mia40 to trap incoming preproteins efficiently at the trans-site of the TOM complex made Mia40 the superior oxidoreductase to mediate the translocation and oxidation of IMS proteins.”

*4) Several papers have specified the cysteine residues of substrates in close proximity of the MISS/ITS signal for Mia40 recognition and showed the import deficiencies of substrates lacking this Cys residue. The in vivo accumulation of these proteins indicates, however, that this "recognition" defect may be bypassed by different means (either by use of other Cys residues or by folding that is not fully dependent on oxidation).*

It would be good to have a WB to see the levels of Erv1 overexpression. Is it possible that substrate proteins form some other intermolecular disulfide bonds that do not involve the CPC motif of Mia40, i.e. with Erv1? Additionally, the possibility of substrate binding by Mia40-SPS via a disulfide bond should be tested in the presence of chemical reductant.

We now added the requested Western blot data as novel Figure 6. Overexpression of Erv1 does not increase the levels of Mia40 substrate in the Mia40-SPS mutant, but rather severely diminishes it. This is consistent with the import experiments into Mia40-SPS mitochondria in which Erv1 was overexpressed. As shown in Figure 4, these mitochondria import Atp23 and Cmc1 rather poorly. Previous studies by Tokatlidis and coworkers already suggested that Erv1 and Mia40 substrates compete for the same binding site on Mia40. Our observations confirm this hypothesis and it appears that overexpression of Erv1 blocks the substrate-binding site in the Mia40-SPS mutant. This effect was obvious for the Mia40 substrates Atp23, Cmc1 and Mrp10 (novel Figure 6).

Furthermore, we performed Western blotting experiments with mitochondrial extracts of the Mia40-SPS mutant in the absence or presence of diamide with or without overexpression of Erv1. As shown in the novel Figure 3, even under these very oxidizing conditions, no adducts are visible on the Mia40-SPS mutant, whereas a very large fraction of Mia40 is engaged in mixed disulfides under the same conditions. From this we conclude that the substrate binding to the Mia40-SPS mutant occurs via the hydrophobic binding cleft rather than via disulfide linkages.

In addition, we added a novel set of pull down experiments in which we tested a potential disulfide-mediated interaction of Cmc1 and Tim9 with Mia40 and Erv1 after import into wild type, Mia40-SPS and Mia40-STOP mitochondria. Whereas both proteins are efficiently precipitated with Mia40, no interactions were observed with Mia40-SPS, Mia40-STOP or Erv1. Even if Erv1 was overexpressed in the Mia40-SPS mutant, only very small amounts of Cmc1 and Tim9 were recovered with Erv1-specific antibodies. These experiments are now shown as novel Figure 5—figure supplement 3.

Moreover, we performed immunoprecipitation experiments with Mia40-SPS-containing mitochondria that were pretreated with 5 mM DTT. As shown in the novel Figure 5, radiolabeled Tim9 and Cmc1 were efficiently bound to Mia40 via disulfide linkage. In contrast, no disulfide-linked adducts could be found with the Mia40-SPS mutant, despite the presence of reductant. This supports our conclusion that the binding to the hydrophobic binding pocket is necessary and sufficient for Mia40-mediated protein import into the IMS.

5) The authors did not provide a sufficient explanation concerning the action of diamide. One can assume that diamide should in fact improve the overall oxidative folding/mitochondrial accumulation of substrates; is this the case? Diamide may or may not improve the substrate binding to Mia40-wt or Mia40-SPS? The answer to this question would be important to specify the stage diamide works. Finally, does diamide influence Mia40 binding to Erv1 or substrate binding to Erv1? The mechanism for diamide should be proposed based on the experimental data.

Diamide is a chemical oxidizer that induces the formation of disulfide bonds. As shown in the novel Figure 3, diamide increases the amount of disulfide-linked substrates on Mia40. However, the CPC motif of Mia40 is essential for this interaction since diamide does not lead to the trapping of interaction partners on Mia40-SPS. Diamide does also not induce the trapping of proteins on Erv1. We conclude that diamide accelerates the disulfide formation in the proteins following Mia40-SPS mediated import.

We also tested in more depth the redox state of proteins imported into the Mia40-SPS mutant. As shown in the novel Figure 6, the steady state levels of Tim10 are oxidized in the mutant, indicating that even in the absence of a functional CPC motif, proteins are oxidized. Moreover, we added the data of a pulse-chase experiment (novel Figure 6) where we analyzed the oxidation of Cox19-HA in wild type and *mia40-4* cells. This also shows that the Mia40- SPS does not improve the oxidation rate of Cox19-HA. This provides additional evidence that Mia40-SPS does not serve as oxidoreductase here.

*6) The import experiments presented in Figure 4 should be described more systematically in the text. The graphs (Figure 4) look very convincing but the information on the number of experimental repetitions taken for the quantification is missing in the Figure Legend. The explanation given in a separate document is not sufficient. Quantifications and average from multiple experiments should be presented, including SEM optimally. Alternatively, it should be clearly stated that one experiment was quantified, and in this case the experiment should be presented in full, not only in the form of bars.*

*What is the purpose of Figure 4? On the one hand there is no WT control to compare with. On the other hand, the quantification of Tim9 import in SPS, FE and WT is depicted in Figure 4. If the authors want to show quantification of kinetics, they should include a WT or otherwise remove the panel.*

*The text does not mention the number of replicates used for the quantitations in Figure 4 and Figure 7 as well as for the growth curves in Figure 3 that appear to have error bars.*

We followed the suggestion of the referee and removed the former panels 4F and 4H. For the quantification shown in the study (Figure 3 and Figure 7), we now indicated the number of replicates (n = 4) and now also provide SD or SEM values as requested.